# Detailed Characterization of Small Extracellular Vesicles from Different Cell Types Based on Tetraspanin Composition by ExoView R100 Platform

**DOI:** 10.3390/ijms23158544

**Published:** 2022-08-01

**Authors:** Kai Breitwieser, Leon F. Koch, Tobias Tertel, Eva Proestler, Luisa D. Burgers, Christoph Lipps, James Adjaye, Robert Fürst, Bernd Giebel, Meike J. Saul

**Affiliations:** 1Fachbereich Biologie, Technische Universität Darmstadt, 64287 Darmstadt, Germany; breitwieser@bio.tu-darmstadt.de (K.B.); leon.koch@tu-darmstadt.de (L.F.K.); ewolf@bio.tu-darmstadt.de (E.P.); 2Institute for Transfusion Medicine, University Hospital Essen, University of Duisburg-Essen, 45122 Essen, Germany; tobias.tertel@uk-essen.de (T.T.); bernd.giebel@uk-essen.de (B.G.); 3Institute of Pharmaceutical Biology, Goethe University Frankfurt, 60438 Frankfurt am Main, Germany; burgers@em.uni-frankfurt.de (L.D.B.); fuerst@em.uni-frankfurt.de (R.F.); 4Department of Internal Medicine I, Cardiology, Justus-Liebig-University Gießen, 35392 Gießen, Germany; christoph.lipps@gmail.com; 5Institute for Stem Cell Research and Regenerative Medicine, Universitätsklinikum Düsseldorf, 40225 Düsseldorf, Germany; james.adjaye@med.uni-duesseldorf.de

**Keywords:** small extracellular vesicles, phenotype, tetraspanins, characterization

## Abstract

Small extracellular vesicles (sEV) hold enormous potential as biomarkers, drug carriers, and therapeutic agents. However, due to previous limitations in the phenotypic characterization of sEV at the single vesicle level, knowledge of cell type-specific sEV signatures remains sparse. With the introduction of next-generation sEV analysis devices, such as the single-particle interferometric reflectance imaging sensor (SP-IRIS)-based ExoView R100 platform, single sEV analyses are now possible. While the tetraspanins CD9, CD63, and CD81 were generally considered pan-sEV markers, it became clear that sEV of different cell types contain several combinations and amounts of these proteins on their surfaces. To gain better insight into the complexity and heterogeneity of sEV, we used the ExoView R100 platform to analyze the CD9/CD63/CD81 phenotype of sEV released by different cell types at a single sEV level. We demonstrated that these surface markers are sufficient to distinguish cell-type-specific sEV phenotypes. Furthermore, we recognized that tetraspanin composition in some sEV populations does not follow a random pattern. Notably, the tetraspanin distribution of sEV derived from mesenchymal stem cells (MSCs) alters depending on cell culture conditions. Overall, our data provide an overview of the cell-specific characteristics of sEV populations, which will increase the understanding of sEV physiology and improve the development of new sEV-based therapeutic approaches.

## 1. Introduction

Previously, it was assumed that cells without direct cellular contacts communicated mainly through soluble molecules such as growth factors, chemokines, and hormones. In the meantime, our understanding of intercellular communication has changed fundamentally. Cells can communicate with neighbouring or distant cells by secreting extracellular vesicles in a controlled manner [1]. They release a heterogeneous group of small extracellular vesicles (sEV, <200 nm in diameter) of endosomal and plasma membrane origin, termed either exosomes or microvesicles (MV), respectively [2,3]. Exosomes are derivatives of the endosomal system and correspond to intraluminal vesicles (ILVs) of multivesicular bodies (MVB). MVBs fuse with the plasma membrane and release their ILVs as exosomes into the extracellular environment. In contrast, MVs bud off of the plasma membrane [1]. Initially, sEV were considered as vehicles for cellular waste disposal [4]. With the discovery that exosomes from B lymphocytes can trigger the activation of T lymphocytes [5], it became clear that sEV have regulatory functions for intercellular communication. They carry a unique composition of macromolecules such as lipids, proteins, and nucleic acids that depend on the characteristics of their parent cells. It is assumed that upon interaction with their target cells, sEV are internalized by recipient cells, and as a consequence of the sEV uptake, alter their cellular behaviour [6,7].

The sEV envelope consists of a lipid bilayer that integrates a variety of surface markers. These surface markers include tetraspanins, integrins, proteoglycans, and other transmembrane and membrane associated proteins. Changes in their composition can influence the activity and function of sEV [6,8]. Thus, sEV are a complex collection of different biomolecules whose composition reflects the biological properties of their parent cells. Therefore, sEV have attracted much attention as therapeutic agents or biomarkers for a particular disease [9,10,11,12]. To make progress in this field, knowledge of the different sEV phenotypes is of great value for developing appropriate therapeutic approaches. A comprehensive characterization of the surface composition at a single sEV level is still lacking due to the technical challenges of analyzing small vesicles [13].

With the majority of sEV being between 30 to 100 nm in diameter [14], sEV are too small to be analyzed by conventional light and flow cytometry technologies [13]. Consequently, sEV were mainly characterized by electron microscopy, usually by transmission electron microscopy (TEM), as has been the case in the studies of reticulocytes and B lymphocytes mentioned above [5,15,16]. In the last decade, knowledge about sEV has mainly been gained through studies of sEV samples obtained with different centrifugation techniques. These isolated sEV were analyzed as homogenates with molecular technologies such as western blot or with different omic approaches [17,18]. The first technology considered to enable sEV analyses at the single vesicle level was nanoparticle tracking analysis (NTA). In 2011, this technology was introduced to quantify and measure exosome size [19,20]. However, the information obtained with conventional NTA analyses can be misleading, because it lacks the ability to differentiate between EVs and non-vesicular contaminants, such as protein aggregates [21]. 

So far, an overview of the different characteristics of individual sEV populations has been lacking. To this day, studies have characterized sEV either by their size, morphology, surface markers, or, when possible, by their content [22]. In an attempt to standardize sEV research, the minimal information for studies of extracellular vesicles 2018 (MISEV2018) guidelines recommend including information on the physical properties of sEV, their biochemical composition, and a description of the cellular origin and isolation methods [2]. New technologies have been developed that allow a multiplex expression analysis of surface proteins at a single sEV resolution [23,24,25]. Since then, a controversial debate has begun about which technology is the best method for characterizing sEV populations.

Overall, these multiplex techniques are either protein-oriented, in which case a specific protein is required for detection, or size-oriented, in which case a sufficient size is needed for the analysis. Single-particle flow cytometry, such as image flow cytometry (IFCM), can analyze single sEV subpopulations by their surface protein profile, such as tetraspanins, at a high throughput rate [21,26,27]. However, it comes with a theoretical detection limit of around 80 nm for sEV [28]. 

On the other hand, single-particle interferometric reflectance imaging sensor- (SP-IRIS) based analysis coupled with immunofluorescence staining is protein-capture oriented and allows nearly true single event detection. It allows the quantification and size analysis of sEV captured by antibodies linked to specific microchips. By combining the method with fluorescence light microscopy, the SP-IRIS based ExoView R100 platform enables simultaneous phenotypic analyses of captured sEV with established fluorochrome-conjugated antibodies [23,29]. Comparable to flow cytometry, this system allows a direct analysis of sEV in biofluids. This fact is a considerable advantage over other analytical methods as it avoids artificial enrichment of the sEV population during the purification process [30].

A recently published study describes the advantages and disadvantages of both methods [28]. While flow cytometry may be more appropriate for high throughput needs, immunocapture likely provides a more holistic protein expression profile due to its ability to visualize smaller particles and detect targets with low expression levels [28,29,31]. Interestingly, both methods seem to affect the determined tetraspanin profile, which is caused by a variety of technical differences that amplify the background signal. Therefore, it is important to consider these limitations when deciding on a characterization method [28]. 

Overall, tetraspanins CD9, CD63, and CD81 are generally considered classical sEV markers [32,33]. In recent years, several studies have shown that sEV from different cell types have different combinations and amounts of these proteins on their surface [34,35]. We supported these findings with the characterization of the sEV phenotype from non-small cell lung cancer (NSCLC) cells [36]. SEV populations of the NSCLC subtypes of adeno- and squamous cell carcinoma exhibit the distinct expression and colocalization of CD9, CD63, and CD81 at a single sEV level. We further demonstrated that among themselves, individual sEV populations are highly heterogeneous [36].

To gain better insight into the complexity and heterogeneity of sEV, we have extended these analyses and created a library of sEV phenotypes secreted from different cell types, including therapeutically relevant sEV from mesenchymal stem/stromal cells (MSCs) and endothelial cells. Our study provides a unique overview of the cell-specific characteristics of sEV populations based on their CD9/CD63/CD81 phenotypes analysed by the ExoView R100 platform.

## 2. Results

An extensive characterization of individual sEV populations of different cell types has yet to be performed, given that appropriate characterization methods have only recently been developed. Therefore, we aimed to characterize sEV populations from minimally processed samples of different cell types and generate a comprehensive overview in order to provide a stepping stone for a future sEV phenotype library. For this approach, we used the ExoView R100 platform. We immunologically immobilized sEV by the membrane proteins CD9, CD63, and CD81 on a microarray chip. The vesicles were counted and sized by light interference (down to 50 nm). In addition, we performed immunofluorescence staining for CD9, CD63, and CD81 to determine the tetraspanin composition at a single vesicle level. Notably, vesicles with a sufficient fluorescence signal were detected irrespective of their determined size by SP-IRIS. Furthermore, the antigen-specific sEV characterization technique of the ExoView R100 platform discriminates between sEV and contaminants, such as bovine-specific sEV.

Overall, we obtained a multidimensional data set from each experiment, which we describe in the following text and summarize in Appendix A.

### 2.1. Organ-Specific Heterogeneity of Fibroblast-Derived sEV Based on the Distribution of CD9, CD63, and CD81

Fibroblasts contribute to the formation and turnover of extracellular matrix components and are essential to the structural integrity of most tissues [37,38]. Over the last years, it has been recognized that fibroblasts display a heterogeneous phenotype across all organs [38]. With this in mind, we aimed to unravel whether the tetraspanin composition of sEV populations reflects this organ-specific heterogeneity of the fibroblasts.

Firstly, we analyzed the size distribution of cardiac fibroblast-derived sEV (Figure 1A). The sEV captured with anti-CD63 and -CD81 antibodies revealed a size of 50 to 60 nm in diameter, while with diameters of 50 to 130 nm anti-CD9 antibody captured EVs appeared more variable in size (Figure 1A). Generally, all three antibody capture spots were equally loaded with vesicles (34.5%, 33.0%, and 32.6%, respectively). Interestingly, a relatively low number of CD9-positive sEV were found on the CD9 antibody capture spot compared with CD63- and CD81-positive sEV (Figure 1B), which indicates that CD9 is mainly localized on the same vesicle with the tetraspanins CD63 and CD81. 56.6% of CD9 positive sEV were positive for all three tetraspanins, and only 6.4% of these vesicles were single positive for CD9. 25.8% of CD9 positive sEV had an occurrence with CD81 and only 11.2% were positive for CD63 and CD9 (Figure 1C). Next, we analyzed sEV from foreskin fibroblasts. The average diameter of these sEV was approximately 55 nm (Figure 1D). Interestingly, the particle distribution of foreskin fibroblast-derived sEV on the three antibody capture spots was comparable to the distribution of cardiac fibroblast sEV (Figure 1E). 31.2% of the sEV were captured by the anti-CD63 antibody, 37.8% of all sEV were bound to the anti-CD81 antibody, and 31% by the anti-CD9 antibody, respectively. The tetraspanin localization pattern was also similar to the tetraspanin composition of cardiac fibroblast sEV (Figure 1F). In detail, 63.4% of CD9 positive sEV revealed also a signal for CD81, CD63, and CD9. 22.7% of CD9 positive foreskin fibroblast-derived sEV were also positive for CD81, and 9.1% showed a localization with CD9 and CD63 on the same vesicle. Only 4.9% of the CD9 positive sEV were single positive for CD9.

Following this, we characterized the tetraspanin profile of the sEV derived from the human lung fibroblast cell line HFL1. The SP-IRIS measurements of the captured sEV showed a homogenous size distribution between CD9, CD63, and CD81 captured sEV, mostly ranging between 50 and 75 nm (Figure 1G). Fluorescent tetraspanin staining of the antibody captured sEV showed that the vesicles derived from HFL1 cells also had a similar but more divergent sEV distribution compared to the sEV population of the other fibroblasts (CD9: 26.4%; CD63: 38.5%; CD81: 35.1%) (Figure 1H). Lastly, we investigated the tetraspanin composition on each individual antibody spot (Figure 1I). 25.4% of all CD9 positive HFL1-derived sEV showed no localization on the same vesicle, 18.6% were positive for CD81 and 3.8% had an occurrence with CD63. The majority of CD9 positive sEV (52.2%) was stained triple positive for all tetraspanins. Moreover, 55.3% of all sEV captured by the ani-CD81 antibody were positive for CD9, CD63, and CD81. 18.5% of CD81 positive sEV localized with CD63 on one sEV and 21.1% were positive for CD9. Only 5.2% of CD81 positive sEV did not localize with the other two tetraspanins on the same vesicle. The anti-CD63 antibody captured sEV showed a unique pattern that deviates from the two previous patterns, with 42% of CD63 positive lung fibroblast sEV not localized with CD81 and/or CD9 on one sEV. In contrast, 32.9% localized with both CD81 and CD9. 18.1% of CD63 positive sEV had an occurrence with CD81 and 7% with CD9.

Overall, the distribution of CD9, CD63, and CD81 captured fibroblast sEV was balanced, with approximately 33%, respectively. Of note, anti-CD81 capture spots showed a tetraspanin phenotype consistent through all three examined fibroblast-derived sEV. However, it was striking that tetraspanin composition was similar in cardiac and foreskin fibroblasts but differed in localization pattern compared with lung fibroblasts. In particular, anti-CD63 and -CD9 captured sEV showed organ-specific differences in their tetraspanin composition.

### 2.2. Cancer Cell-Derived sEV Populations Can Be Distinguished by the Abundance of the Tetraspanins CD9, CD63, and CD81

In a previous study, we recognized that sEV derived from different NSCLC subtypes did not differ in size distribution, but showed a different tetraspanin composition [36]. This observation motivated us to analyze the sEV populations of other cancer cell lines. As a first step, we analyzed the tetraspanin colocalization pattern of the cervical cancer cell line HeLa. The overall size distribution was between 50 and 120 nm (Figure 2A). The particle fractions on the CD9, CD63, and CD81 antibody capture spots were 36.2%, 21% and 42.8%, respectively (Figure 2B). We next analyzed the abundance of the tetraspanins on the sEV level (Figure 2C). 56.7% of CD9 positive HeLa cell-derived sEV were negative for CD81 and CD63. 17.3% showed a CD81 expression, 8.5% revealed a colocalization on the same vesicle with CD63, and 17.6% were positive for all three tetraspanins. When captured by CD81 antibody, 38.8% of HeLa cell-derived sEV showed colocalization with CD9, 16.7% were positive for CD63, and 26.3% of all CD81 positive vesicles localized on the same sEV with all three tetraspanins. 18.1% were single positive for CD81. On the CD63 spot, 50.7% of the captured sEV showed a CD63 and CD9 localization on the same vesicle. 16.3% colocalized with CD81 on one sEV, 17.1% were single positive for CD63, and only 15.8% were on the same vesicles with all tetraspanins. Overall, these results suggest that the HeLa cell-derived sEV population exhibits a cell specific tetraspanin composition with a high amount of CD9 single positive sEV. We then investigated the tetraspanin composition of sEV derived from the hepatoma cell line HepG2. The size of HepG2 cell-derived sEV captured by anti-CD63 and -CD9 antibodies ranged from 50 to 75 nm, whereas sEV captured by the anti-CD81 antibody revealed a size distribution up to 110 nm (Figure 2D). Most sEV were bound by the anti-CD63 and -CD9 antibody (39.1% and 52.7% of total sEV, respectively). The particle number of the CD81 antibody capture spot was low (8.2%) and comparable to the background of the IgG control (Figure 2E). The tetraspanin abundance of HepG2 cell-derived sEV showed a distinct composition (Figure 2F).

On the CD9 antibody spot, the majority (94.3%) of the captured vesicles were single positive for CD9. Only 3.2% revealed localization of CD81 and CD9 on one vesicle, and only 2.5% were positive for CD63 and CD9. Of note, we observed no localization of all three tetraspanins on this spot. On the contrary, 76.4% of all CD81 positive HepG2 cell-derived sEV showed a CD9 localization on the same sEV. 21.0% were single positive, and only 2.6% had an occurrence of CD81 and CD63.

Tetraspanin abundance of CD63-positive HepG2 cell-derived sEV showed that 56% of these vesicles do not localize with the other tetraspanins. 36.5% were positive for both CD63 and CD9, 7.3% had an occurrence with CD63 and CD81, and only 0.2% were positive for all three tetraspanins. These results indicate that the sEV population of HepG2 is negative for the colocalization of all three tetraspanins: CD9, CD63, and CD81, which seems to be unique to this cell line.

We next analyzed the sEV population of the NSCLC adenocarcinoma cell line H1650. The average size of H1650-derived sEV ranged between 55 to 110 nm in diameter (Figure 2G). Most H1650 cell-derived sEV were bound by anti-CD9 antibody (71% of total sEV). Only 17.3% of all sEV were CD81 positive, and 11.7% were CD63 positive (Figure 2H). Next, we analyzed the localization of CD9, CD63, and CD81 on the same vesicle (Figure 2I). 45.8% of all CD9 positive H1650 cell-derived sEV showed no localization with the other tetraspanins. 28.9% of these vesicles were positive for CD81, 12.6% were positive for CD63 and CD9, and 12.6% had an occurrence with CD9, CD63, and CD81. This tetraspanin pattern stands in contrast to the composition of CD81 positive sEV. Here, 73.9% of vesicles were positive for CD81 and CD9, 21.9% showed localization of CD63, CD81, and CD9 on one sEV, 2.1% showed no occurrence with CD63 or CD81, and, finally, 2.2% of all CD81 positive sEV colocalize with CD63 on the same vesicle. On the CD63 capture spot, 51% of H1650 cell-derived sEV revealed a localization with CD9 on one sEV. 26.6% had an occurrence with all three tetraspanins, whereas 16.1% of all sEV showed no localization with CD9 and CD81. 6.3% of all CD63 positive sEV were also positive for CD81. Following this analysis, we compared our results of the H1650 sEV population with our previous study in which the sEV tetraspanin composition from the adenocarcinoma cell line A549 and the squamous cell carcinoma cell line 2106T were analyzed [36]. We found that the tetraspanin abundance of CD9 positive sEV was similar in all NSCLC cell lines. With slight differences, the tetraspanin composition of CD63 positive sEV was alike in both lung adenocarcinoma cell-derived sEV populations, but not in 2106T cell-derived sEV. Finally, the tetraspanin composition of the CD81 positive sEV populations of A549- and 2106T cell-derived sEV were comparable, which indicates that the composition was different for H1650 cell-derived sEV. 

Finally, we analyzed the tetraspanin composition of sEV derived from the neuroblastoma cell line SH-SY5Y. The size distribution of SH-SY5Y cell-derived sEV was homogenous throughout all capture antibodies ranging from 50 to 80 nm in average diameter (Figure 2J). Most SH-SY5Y cell-derived sEV were captured by CD81 and CD9 antibodies (46.1% and 30.5%, respectively), while 23.3% of all vesicles were captured by CD63 (Figure 2K). We next analyzed the distribution of CD9, CD63, and CD81 on one sEV (Figure 2L). 29.5% of all CD9 positive SH-SY5Y cell-derived sEV revealed no localization with CD81 or CD63 on the same vesicle. 34.3% of all CD9 positive sEV had an occurrence with all three tetraspanins. 26.8% of sEV captured by anti-CD9 antibody were positive for CD81 and 9.4% localized with CD63 and CD9 on one sEV. The tetraspanin composition of sEV bound to the anti-CD81 antibody appeared to be distributed roughly equal between all possible tetraspanin combinations. 28.5% of CD81 positive sEV revealed a localization with CD63 and CD9. 28.3% were positive for CD63 and CD81, and 20.4% showed a CD81 and CD9 occurrence on the same sEV. Finally, 22.8% of CD81 positive sEV did not localize with CD9 or CD63 on one sEV. On the CD63 spot, 43.8% of SH-SY5Y cell-derived sEV were positive for CD81, 29.8% had an occurrence for all three tetraspanins, and 17.3% were positive for CD9, and 9.2% were single positive for CD63. 

Overall, we demonstrated that the surface markers CD9, CD63, and CD81 are sufficient to distinguish between sEV populations derived from different cancer cells. In particular, the tetraspanin phenotype on sEV derived from HepG2 cells was unique compared to all analyzed cell types. Here, nearly no CD81 positive sEV were observed. The majority of the HepG2 derived sEV population was CD9 and CD63 single positive sEV, respectively.

### 2.3. Similar Tetraspanin Distribution of Endothelial Cell-Derived sEV

Here, we examined the abundance of CD9, CD63, and CD81 on sEV derived from two types of endothelial cells: human microvascular endothelial cells (HMEC-1) and primary human umbilical vein endothelial cells (HUVEC). The size distribution of both endothelial cells was similar and ranged from 50 to 80 nm in diameter (Figure 3A,D). In both cell lines, most sEV were captured by the anti-CD9 antibody (HMEC-1: 49.8%; HUVEC: 47.3%) followed by the anti-CD81 antibody (HMEC-1: 28.3%; HUVEC: 32.3%) and CD63 (HMEC-1: 21.8%; HUVEC: 20.4%) (Figure 3B,E). Next, we analyzed the localization of CD9, CD63, and CD81 on one vesicle level (Figure 3C,F). In general, the tetraspanin pattern of both endothelial cells was very similar, with slight differences on the CD63 and CD81 antibody capture spot.

In detail, in 37.0% of HMEC-1-derived CD9 positive sEV, CD9, CD63, and CD81 localized on one vesicle. 24.0% were positive for CD81, 29.0% revealed no occurrence with other tetraspanins, and 10.1% localized with CD63 on one sEV. In 33.6% of HUVEC-derived CD9 positive sEV, CD9, CD63, and CD81 were localized on the same vesicle. 31.7% revealed an occurrence with CD81, 26.9% did not localize with the other tetraspanins, and 7.8% were positive for CD63. On the CD81 antibody spot, 56.8% of HMEC-1-derived sEV were positive for all three tetraspanins, while 46.7% of HUVEC-derived sEV colocalized with CD9, CD63, and CD81 on one sEV. 46.4% of CD81 positive HUVEC-derived sEV localized with CD9 on one sEV, and only 37.6% of CD81 positive HMEC-1 sEV were also positive for CD9. About 4% of both CD81 positive endothelial cell-derived sEV populations localized with CD63 on the same sEV. Finally, no occurrence of the other tetraspanins was observed in 1.5% or 2.6% of CD81-positive vesicles. When captured by the anti-CD63 antibody, 59.2% of all HMEC-1-derived sEV localized with CD9, CD63, and CD81 on one sEV. 21.6% of all CD63 positive sEV had no occurrence with CD9 and CD81. 16.4% were positive for CD63 and CD9, and only 2.8% were positive for CD63 and CD81. In 46.1% of CD63 positive HUVEC-derived sEV, a localization with the other two tetraspanins on one sEV was observed. 39.0% of these vesicles were individually positive for CD63, while 10.8% showed localization with CD9 and 4.2% showed localization with CD81 on the same vesicle. A comparison of the sEV population derived from two types of endothelial cells revealed a comparable CD9/CD63/CD81 phenotype. However, we observed slight differences in the tetraspanin composition, especially on CD63 positive sEV.

### 2.4. Tetraspanin Abundance of Stem Cell-Derived sEV Can Be Altered by Cell Culture Conditions

We analyzed the composition of CD9, CD63, and CD81 on sEV derived from two types of stem cells. Due to their well-known therapeutic value, we have chosen to characterize the sEV phenotype of MSCs and induced pluripotent stem hiPS cells [39,40].

The first step aimed to characterize sEV populations of hiPS cells. For this purpose, we have chosen the hiPS cell line A4, derived from healthy neonatal foreskin fibroblasts. The average diameter of hiPS cell-derived sEV revealed 60 nm, although individual vesicles were larger, with diameters > 120 nm (Figure 4A). All antibody capture spots were on average equally loaded (CD9: 33.6%, CD63: 28.8% CD81: 37.6%) (Figure 4B). Furthermore, we analyzed the localization of each tetraspanin on one sEV level. Generally, we recognized that most captured sEV were positive for all three tetraspanins, which differs from MSC-derived sEV (Figure 4C). On the CD63 spot, we observed 61.4% triple positive sEV, 15.8% were positive for CD63 and CD9, 11.6% were positive for CD63 and CD81, and 11.3% were positive only for CD63. Similarly, when captured by anti-CD81 antibody the majority of vesicles (53.6%) were triple positive, followed by CD81/CD9 colocalizing sEV (34.7%). 7.6% of sEV were positive for both CD81 and CD63, and only 4.0% of CD81 positive sEV had no occurrence with either CD63 or CD9. 

A comparable tetraspanin composition was observed in sEV captured with an anti-CD9 antibody. Here, 55.9% of CD9 positive sEV were positive for all three tetraspanins. 27.3% of CD9 positive sEV showed localization with CD81 on the same sEV, 10% revealed a CD63 occurrence, and 6.7% of all CD9 positive sEV did not localize with other tetraspanins on the same vesicle.

Additionally, we also analyzed sEV derived from the hiPS cell line CO2. This cell line was derived from healthy dermal fibroblasts. SEV size was more homogenous compared to A4-derived sEV and generally ranged between 50 nm and 75 nm (Figure 4D). Most sEV were captured by the anti-CD81 antibody (37.9%), followed by the anti-CD63 antibody (33.3%) and anti-CD9 antibody (28.7%; Figure 4E). The tetraspanin composition profile of CO2-derived sEV showed a similar pattern when compared to A4-derived sEV (Figure 4F). Here, triple positive sEV also made up the majority of colocalizing sEV, irrespective of the capture spot. The CD81 colocalization spot profile between A4- and CO2-derived sEV appeared very similar. The composition of the CO2-derived sEV population captured by the anti-CD9 antibody also showed a pattern comparable to the A4-derived population. However, the CD81/CD9 fraction was more pronounced at the cost of a triple positive sEV fraction. The biggest differences between the two hiPS cell-derived sEV populations were observed at the CD63 antibody spot. CO2-derived sEV captured by anti-CD63 antibody were mostly positive for both CD81 and CD9 (41.8%), followed by single positive sEV (21%) and CD63/CD9 localizing sEV (23.6%). Only 13.5% of CD63 captured sEV localized with CD81 on the same sEV.

Next, we analyzed sEV derived from MSCs that were cultured in the presence of human platelet lysate (hPL). Here, we observed that most sEV ranged between 50 to 75 nm in diameter (Figure 4G). Most MSC-derived sEV were bound to anti-CD63 and -CD9 antibodies (45% and 38.2% of total sEV). Only 16.7% of all sEV were CD81 positive (Figure 4H). Next, we analyzed the abundance of CD9, CD63, and CD81 (Figure 4I). 63.1% of CD63 positive MSC-derived sEV showed no localization with the other tetraspanins on one sEV. Only 10.3% had an occurrence with CD81. 11.1% were also positive for CD9, while 15.5% were positive for all three tetraspanins. CD9 captured MSC-derived sEV revealed a similar tetraspanin composition. 78.1% were only positive for CD9, 2.8% localized with CD63 on the same vesicle, 7.6% had an occurrence with CD81, and 11.6% were positive for all three tetraspanins (Figure 4I). Interestingly, the low amount of CD81 positive sEV revealed a high localization rate on the same vesicle with both CD63 (19.4%), CD9 (29.4%), or both tetraspanins together (45.9%). Only 5.2% of the CD81 positive sEV had no occurrence with the other tetraspanins (Figure 4I). 

Additionally, we investigated how supplementation of media with hPL affects the tetraspanin profile. Therefore, we cultivated MSCs in media supplemented with FCS instead of hPL. SP-IRIS measurements of FCS cultivated MSC-derived sEV showed similarly distributed sEV sizes compared to hPL cultivated MSC-derived sEV (Figure 4J). In contrast, FCS cultivated MSC-derived sEV revealed a more prominent CD81 positive sEV fraction resulting in a more balanced distribution (CD9: 31.2%; CD63: 35.2%; CD81: 33.6%; of total sEV, respectively). FCS cultivated MSC-derived sEV also showed a distinct tetraspanin abundance when compared with hPL cultivated MSC-derived sEV (Figure 4L). On the CD63 spot, 32.4% of sEV did not localize with one of the other two tetraspanins on the same sEV. Most CD63 antibody captured sEV (35.6%) localized with CD81 on one vesicle. 23% of all CD63 antibody captured sEV were positive for all three tetraspanins. Only 9.1% were positive for CD9. 35.4% of FCS cultivated MSC-derived sEV were captured by anti-CD81 antibody and localized with CD63, while 21.2% localized with CD9 on the same vesicle. 33.5% of all CD81 captured sEV were positive for CD9 and CD63, while only 10% did not show colocalization. The CD9 spot was mostly dominated by single positive sEV (49.2%). 29.2% of CD9 antibody captured sEV had an occurrence with both CD81 and CD63. 18.4% were positive for CD81 and only 3.2% of CD9 captured sEV localized with CD63 on the same vesicle.

It becomes apparent from this data that the tetraspanin phenotype between two types of iPS cells is comparable, but different to MSC-derived sEV. Of note, we recognized that the phenotype of sEV might change depending on cell culture conditions.

### 2.5. The Tetraspanin Distribution of the Different sEV Populations Is Mostly Randomly Distributed

Finally, we analyzed the randomness (Appendix A) of the observed tetraspanin distribution. We aimed to determine whether specific tetraspanin combinations were preferentially formed. First, we investigated the distribution of hPL cultivated MSC-derived sEV (Figure 5A,B). Interestingly, the CD63 antibody captured sEV population showed deviations from the predicted distribution. Non-colocalizing (63.1% measured; 54.5% predicted) and triple positive sEV (15.5% measured; 6.9% predicted) were over-represented. Also, the CD9 captured sEV population revealed an over-representation of single positive (78.1% measured; 69.3% predicted) and triple positive sEV (11.6% measured; 2.7% predicted). In contrast, CD81 captured sEV revealed a randomly distributed tetraspanin composition. 

Interestingly, similar trends were observed when examining the randomness of sEV colocalization of MSCs cultured in FCS (Appendix A). The CD63 captured sEV population showed a slight over-representation of single positive (measured: 32.4%; predicted: 28.2%) and triple positive sEV (measured: 23%; predicted: 18.8%), while the CD81 positive sEV population followed the predicted random distribution. The CD9 antibody captured sEV population again showed an over-representation of single positive (measured: 49.2%; predicted: 35.4%) and triple positive sEV (measured: 29.2; predicted: 15.5%). Both hiPS cell-derived sEV populations revealed a tetraspanin localization pattern on one sEV closely resembling the calculated random distribution on the CD81 and CD9 antibody capture spots (Appendix A). In contrast, both sEV populations deviated from the calculated random distribution at the CD63 antibody capture spot. Here, an over-representation of both triple positive (A4: 61.4% measured, 56.3% predicted; CO2: 41.8% measured, 36.2% predicted) and single positive sEV (A4: 11.3% measured, 6.2% predicted; CO2: 21% measured, 15.4% predicted) was revealed. An examination of the fibroblast-derived sEV tetraspanin composition revealed no deviations from randomness, irrespective of tetraspanin antibody capture (Appendix A). Only lung fibroblast-derived sEV appeared to differ from the predicted randomness (Figure 5C,D). In detail, the CD63 captured population had an overrepresentation of single positive sEV (42% measured; 29.4% predicted) as well as triple positive sEV (32.9% measured; 20.4% predicted). Similar discrepancies were found when investigating the CD9 positive sEV population of lung fibroblast-derived sEV. Single positive sEV were over-represented (25.4% measured; 12.9% predicted) as were triple positive sEV (52.2% measured; 39.6% predicted). 

Next, we investigated the randomness of the tetraspanin abundance of sEV derived from cancer cells. HeLa cell-derived sEV showed a slight under-representation of single positive CD63 positive sEV (17.1% measured; 22.7% predicted; Appendix A).

The CD81 capture spots showed distribution patterns resembling the simulated random distribution, and the CD9 spot also showed an over-representation of non-colocalizing (56.7% measured; 48.1% predicted) and triple positive (17.6% measured; 9.1% predicted) sEV. HepG2-derived sEV (Appendix A), as well as H1650-derived sEV (Appendix A), revealed a composition akin to the calculated random distribution. In the case of SH-SY5Y-derived sEV (Appendix A), both CD63- and CD81-positive sEV populations also showed a random distribution. However, sEV captured by the anti-CD9 antibody showed a slight overrepresentation of non-colocalizing and triple positive sEV. Finally, a closer look at endothelial cell-derived sEV uncovered deviations on sEV captured by anti-CD63 antibody derived from both HMEC-1 and HUVEC sEV. The fraction of single positive CD63 sEV was overrepresented (~2-fold higher than random) in both experimental data sets (Figure 5E–H) compared to the simulated random distribution. Additionally, triple positive sEV were also over-represented (HMEC-1: ~1.3-fold; HUVEC: ~1.6-fold). In contrast, the tetraspanin localization patterns on sEV captured by anti-CD81 behaved consistently with the simulated random distribution. When sEV bound to the anti-CD9 antibody, slight deviations from randomness could be observed for both endothelial cell-derived sEV populations. A sole colocalization with CD63 or CD81 was slightly underrepresented in the experimental data, whereas non-colocalizing and triple positive sEV were slightly overrepresented. Both endothelial cell-derived sEV populations showed similarities in their deviations from randomness.

These results indicate that the tetraspanin distribution on sEV does not always follow a random pattern, suggesting that regulatory processes may modulate the CD9/CD63/CD81 phenotype in certain cell types.

## 3. Discussion

SEV have attracted much attention due to their role in cellular communication and as a mediator of pathobiological processes [41,42,43,44]. Since they represent a functional unit, they are also of great importance for clinical and translational applications [45,46]. Therefore, the characterization of sEV is one of the most critical research areas to understand their biological properties, explore their potential for biomarker development, and engineer sEV-based drug delivery to specific target cells [22,47,48]. Despite considerable efforts in this relatively new field of research, our knowledge of different sEV phenotypes remains incomplete due to the technical limitations of sEV characterization. 

In our experimental setup, we chose a combined SP-IRIS-immunofluorescence approach to analyze the phenotype of sEV populations. It is one of the best methods for understanding the surface marker ratios within an sEV population and phenotyping single particles [49]. We would like to point out that only sEV expressing either CD81, CD63, or CD9 can be analyzed in this experimental setup. However, combining all three surface markers should mitigate this limitation of protein capture [28]. Moreover, the reported size determination is probably not consistent with results obtained by other methods such as NTA, since the drying of the preparations causes shrinkage of the vesicles [29]. In addition to this, the platform does not provide a total particle count, but rather an events number detected on each antibody captured spot. 

Here, we provide a comparative analysis of tetraspanin abundance of an assortment of sEV populations. Our findings highlight the cell type-specific tetraspanin heterogeneity for the well-established sEV markers CD9, CD63, and CD81. We expanded our previous cancer cell-derived sEV tetraspanin colocalization analysis [36] by studying further sEV populations from different cancer cell entities. Interestingly, all analyzed cancer sEV populations show diverse and distinct tetraspanin profiles. No common features of cancer sEV could be derived, which reflects the heterogeneity of cancer and the challenges it presents [50,51]. However, this heterogeneity also offers an advantage when it comes to the identification of new biomarkers for cancer therapies, as one can use cancer type-specific characteristics for such approaches. 

We further extended our study to investigate if cell or tissue type-specific tetraspanin composition patterns would be found. During our analysis of three different fibroblast cell lines, we observed that cardiac and foreskin fibroblast-derived sEV featured a very similar CD9/CD63/CD81 phenotype, whereas lung fibroblast-derived sEV showed a pattern that differs widely on the CD63 and CD9 antibody capture spots. These differences might be attributed to cell culture conditions or to fibroblast heterogeneity in general. Fibroblasts play a major role in tissue homeostasis and their function varies depending on tissue type [37]. These results indicate distinct gene expression profiles [38]. Recently, Yeung and colleagues were able to show in a proteomics approach that this heterogeneity also extends to sEV and their cargo [52]. This task-specific heterogeneity might also in part be reflected by the observed sEV tetraspanin profiles. Additionally, we included sEV populations derived from MSCs and hiPS cells, as well as endothelial cell-derived sEV. Here, common tissue-specific tetraspanin features could be observed.. Both endothelial cell-derived sEV populations share commonalities on CD81 and CD9 capture. Similarities between CD81 and CD9 were also observed for both hiPS cell-derived sEV populations. However, MSC-derived sEV differed widely from both hiPS cell-derived sEV populations and featured a unique tetraspanin profile that was dominated by CD9 and CD63 single positive sEV. These features might be crucial to the unique therapeutic properties attributed to MSC-derived sEV in different diseases [39,53]. Despite the therapeutic value of MSC-derived sEV in preclinical studies, the transfer of these findings into a clinical setting requires a large-scale production under reproducible conditions [39,54]. From this point of view, our discovery that the tetraspanin composition of MSC-derived sEV can change depending on cell culture conditions is notable. The composition of the cell medium is critical for developing an MSC-sEV manufacturing process.

In addition, our results also highlight the fact that the immune-affinity enrichment of sEV with antibodies against a tetraspanin should be carefully considered in the experimental design. As we have seen in our study, the entirety of an sEV population may not be detected. This stand in line with other publications recognizing that sEV populations are very heterogeneous, and the classical EV markers might not be sufficient alone to discriminate EV populations [28,34,55].

The differences observed for the lung fibroblast-derived sEV and the unique tetraspanin distribution of the MSC-derived sEV gave us reason to further investigate whether we could detect a pattern that would characterize these deviations. This led us to the calculation of the random distribution which can serve as a reference when analyzing tetraspanin composition on one sEV. Interestingly, both lung fibroblasts and MSC-derived sEV showed aberrations from the calculated random distribution. Additionally, small deviations were observed for other tetraspanin localization patterns. Most strikingly, both endothelial cell-derived sEV populations severely deviated from the random prediction. This might indicate an underlying biogenesis mechanism in this cell type that could be linked to their physiological function. In our small sEV cohort, most of the observed tetraspanin compositions followed a random distribution. This was especially true for the tetraspanin localization observed on the CD81 capture spot, which closely resembled a random distribution in all observed sEV populations. Deviations from the random distribution seem to follow a common pattern, meaning that they vary on the CD63 and CD9 capture spots. On both spots, the non-colocalizing and triple positive subfractions seem to be overrepresented. We hypothesize that in the cells that secrete non-randomly distributed sEV, an underlying protein sorting mechanism could be active, which is in line with previous studies [33,56,57]. Our results suggest that there is a biogenesis mechanism that appears to favor the formation of CD9- or CD63-positive or triple-positive sEV. This fact is particularly interesting since Kowal et al. recognized that dendritic cell-derived vesicles bearing only CD9 or CD81 might not be formed in endosomes (so-called ectosomes). CD63 positive vesicles in combination with CD9 or CD81 might correspond to endosome-derived sEV [34]. These results were supported by a recent publication analysing the spatio-temporal transport of tetraspanins from their initial synthesis in the endoplasmic reticulum to the secretion of sEV [55]. In HeLa cells, CD63 and CD9 are released in small ectosomes formed at the plasma membrane. Exosomes represent a minor vesicle subpopulation that carries CD63 along with other late endosomal markers [55]. However, further research needs to be conducted in order to characterize underlying biogenesis mechanisms and to identify which stimuli or physiological conditions are involved in the activation of these pathways. Moreover, it needs to be investigated whether the heterogeneity of the sEV population corresponds to specific functional properties. Developments in sEV analyses have led to improvements in the quantification and characterization of sEV populations over the last years. While most of the techniques provide information about an sEV population, it is of great interest to have information about the size and molecular characteristics on a single vesicle level of an sEV population [58]. Since such multiplex sEV analyses on a single sEV level are merely getting started, characterization studies of sEV populations are limited. Therefore, the overall aim of this study was to generate a library of different sEV populations derived from various cell types based on the localization pattern of CD9, CD63, and CD81 on the single vesicle level. We conclude that the distribution of these surface markers is sufficient to distinguish cell-type-specific sEV phenotypes and that their composition does not always follow a random pattern. Such information is, from our point of view, crucial for a correct interpretation of sEV-related research studies and the improvement of sEV-based therapeutic approaches. 

We are aware that different sEV characterization methods have been developed at a single sEV level and are currently being discussed in terms of their advantages and disadvantages. Therefore, we consider our study as a starting point for further studies dealing with the characterization of sEV populations on a single sEV level. The results of these studies should be compiled and made available to researchers with a detailed description of the methods. This would help to improve new sEV characterization methods. Furthermore, it would increase the understanding of sEV biology and support the development of sEV-based biomarkers and therapeutic applications.

## 4. Materials and Methods

### 4.1. Cell Lines and Cell Culture

Within the project, different human cell line cells were raised and expanded for obtaining sEV containing conditioned cell culture media. 

The human foreskin fibroblast cell line CI-huFIB (InSCREENeX, Braunschweig, Germany) [59] and the human cardiac fibroblast cell line (huCarFib), derived from primary human ventricular cardiac fibroblast (NHCF-V, Lonza, Basel, Switzerland), were generated by immortalization using as described previously [59]. Both cell lines were expanded on collagen (InSCREENeX, Braunschweig, Germany)-coated plastic in serum-free huFIB medium (InSCREENeX, Braunschweig, Germany) without additional supplements. 

The human lung fibroblast cell line HFL1 (ATCC: CCL-153™) was cultivated in Ham’s F-12K medium (Thermo Fisher Scientific, Waltham, MA, USA) supplemented with 10% heat inactivated fetal calf serum (FCS; Sigma-Aldrich, St. Louis, MI, USA) and 50 mg/mL gentamycin (Merck Millipore, Billerica, MA, USA). 

The human lung adenocarcinoma cell line H1650 (ATCC: CRL-5883™) was cultured in RPMI 1640 medium (Thermo Fisher Scientific, Waltham, MA, USA) with 10% heat-inactivated FCS, 100 U/mL penicillin and 100 μg/mL streptomycin (Carl Roth, Karlsruhe, Germany), 1 mM sodium pyruvate (Thermo Fisher Scientific, Waltham, MA, USA), 10 mM 4-(2-hydroxyethyl)-1-piperazineethanesulfonic acid (HEPES; Carl Roth, Karlsruhe, Germany), and 25 mM glucose (Carl Roth, Karlsruhe, Germany). 

The human epithelial cervix adenocarcinoma cell line HeLa (ATCC: CCL-2™) was expanded in Dulbecco’s Modified Eagle’s Medium (DMEM; Thermo Fisher Scientific, Waltham, USA) with 10% heat inactivated FCS (Sigma-Aldrich, St. Louis, MI, USA), 100 U/mL penicillin, 100 µg/mL streptomycin (Carl Roth, Karlsruhe, Germany) and 1 mM sodium pyruvate (Thermo Fisher Scientific, Waltham, MA, USA). 

The human hepatocellular carcinoma cell line HepG2 (ATCC: HB-8065™) was cultivated in DMEM supplemented with 10% heat-inactivated FCS, 100 U/mL penicillin, 100 µg/mL streptomycin and 1 mM sodium pyruvate. In addition to trypsin digestion, cells were flushed three times through a 20 G hypodermic-needle to separate cell clusters. 

The human epithelial neuroblastoma cell line SH-SY (ATCC: CRL-2266™) was cultured in DMEM (Thermo Fisher Scientific, Waltham, USA) supplemented with 10% heat-inactivated FCS (Sigma-Aldrich, St. Louis, MI, USA) and 50 mg/mL gentamycin (Merck Millipore, Billerica, MA, USA). 

Primary HUVECs were isolated as described previously [60]. HUVECs and the microvascular endothelial cell line HMEC-1 (lot 119223; CDC, Atlanta, GA, USA) were expanded on collagen G (Biochrom, Berlin, Germany)-coated plastic using easy ECGM (PELOBiotech, Planegg, Germany) supplemented with 10% FCS (Biochrom, Berlin, Germany), 100 U/mL penicillin, 100 µg/mL streptomycin (PAN-Biotech, Aidenbach, Germany), 2.5 µg/mL amphotericin B (PAN-Biotech, Aidenbach, Germany) and a supplement mixture (PELOBiotech, Planegg, Germany). HUVECs were cultivated to passage 2 and used for experimental purposes in passage 3. A waiver has been granted for the use of anonymized human material issued by the head of the IRB on 15 September 2021 under the reference number W1/21Fü.

If not stated otherwise, cells were used for experimental purposes up to passage 25 and passaged twice weekly. Therefore, cells were washed briefly with 1× phosphate buffered saline (PBS; Thermo Fisher Scientific, Waltham, MA, USA) and detached with 0.5% Trypsin EDTA (Thermo Fisher Scientific, Waltham, MA, USA) in PBS in the incubator. Digestion was stopped with fresh culture medium and cells were seeded into a new T75 cell culture flask (Greiner Bio-One, Kremsmünster, Austria). All cells were expanded under standard cell culture conditions (37 °C, 5% CO_2_, 98% humidity).

Cells of the human foreskin and dermal fibroblast-derived hiPS cell lines A4 and CO2 were generated as described previously [61,62]. The CO2 cells were maintained on Matrigel^®^ (hESC-qualified; Corning, NY, USA)—coated 6-well plates in mTeSR™ Plus (Stemcell Technologies, Vancouver, BC, Canada). The passaging of cells was performed once a week with 1× PBS with a ratio of 1:6. For the generation of supernatant, the medium was exchanged for fresh medium and cultivated for 24 h before cell passaging took place. Supernatants of A4 cells were kindly provided by Nils Offen (TU Darmstadt). For the generation of supernatant, medium of cells at 80–90% confluency was exchanged with fresh medium and cultured overnight.

Human MSCs were raised from bone marrow samples of healthy donors after informed consent according to the declaration of Helsinki, as described previously [63,64]. All experiments were approved by the ethics committee of the der Medical faculty of the University of Duisburg-Essen to raise MSCs for the extracellular vesicle production (12-5294-BO). Briefly, MSCs were expanded in DMEM low glucose media (Lonza Bioscience, Bases, Switzerland) supplemented with 10% hPL (in-house produced by the lab of B. Giebel) or 10% fetal bovine serum (FBS, Gibco, Thermo Fisher Scientific, Waltham, MA, USA), 100 U/ mL penicillin-streptomycin-L-glutamine (Thermo Fisher Scientific, Waltham, MA, USA) and 5 IU/ mL heparin (Ratiopharm, Ulm, Germany). Media exchanges were performed when MSCs reached 50% confluence. Thereafter, media were exchanged for the sEV preparation every 48 h, until cells reached 80–90% confluence, and then cells were passaged. Harvested media were collected and centrifuged at 2000× *g* for 15 min to remove cellular debris and stored at −80 °C until further processing.

### 4.2. Preparation of Cell Culture Supernatants

For the characterization of CD9, CD63, and CD81 contents on sEV, the CM of the different cells were prepared as follows. If not stated otherwise, cells were passaged, then seeded in their preferred medium into individual wells of given 6-well plates (Greiner Bio-One, Kremsmünster, Austria) and cultured overnight until cellular confluence between 70–80% was achieved. The next day, the CM was harvested and centrifuged at 13,000× *g* for 5 min at room temperature to sediment detached cells and cellular debris. After centrifugation, the supernatant was either stored on ice for later analysis or stored at −80 °C until further use. 

### 4.3. Immunofluorescent Tetraspanin Staining

Microarray chips coated with antibodies against anti-CD9 (HI9a), -CD63 (H5C6) and -CD81(JS-81) (NanoView Biosciences, Boston, MA, USA Product No.: EV-TETRA-C) were pre-scanned according to the manufacturer’s instructions to generate baseline measurements of pre-adhered particles before sample incubation. The cell debris-cleared CM was diluted in incubation solution (NanoView Biosciences, Boston, MA, USA) in order to avoid oversaturation of the chip, and 40 µL were carefully loaded onto the pre-scanned chip and incubated overnight. The incubation was carried out at room temperature in a sealed 24-well plate (Greiner Bio-One, Kremsmünster, Austria, Product No.: 662160) to prevent sample evaporation. The following day, anti-CD9, -CD63 and -CD81 stainings of the captured sEV were performed according to the manufacturers’ instructions. According to the manufacturer, the tetraspanin antibodies applied to the chips and used for fluorescence detection are identical clones. All reagents were stored at 4 °C and were allowed to reach room temperature before use. All incubation steps were performed with gentle shaking (350 rpm on an orbital shaker). First, the microarray chips were submerged in 1000 µL solution A (NanoView Biosciences, Boston, MA, USA). Subsequently, 750 µL of solution A were aspirated and replaced with 750 µL of fresh solution A. This washing procedure was repeated a total of three times with a 5 min incubation in between. In parallel, the antibody detection mixture was prepared as a twofold concentrate by diluting anti-CD9 (CF^®^-488-labeled), -CD63 (CF^®^-647-labeled) and -CD81 (CF^®^-555-labeled) antibodies 1:500 in blocking buffer (antibody and blocking buffer both from NanoView Biosciences, Boston, MA, USA). After the third withdrawal of 750 µL of solution A, 250 µL of the antibody mixture were added and incubated for one hour at room temperature. The 24-well plate was covered with aluminum foil for all subsequent steps to minimize fluorophore bleaching. Unbound antibodies were washed out by adding 500 µL of solution A, thus bringing the total volume back to 1000 µL, and performing three wash cycles with solution A as described above. Excess detergent was removed by washing three times with Solution B (NanoView Biosciences, Boston, MA, USA).

Finally, 750 µL of solution B were aspirated and replaced with 750 µL of Milli-Q water. In order to dry the chip, it was carefully removed from the well with tweezers and placed in a Petri dish filled with Milli-Q water, taking care not to dry out the antibody spots. After carefully swirling the Petri dish, the chip was slowly removed from the Milli-Q bath at a 45° angle to allow the surface tension of the water to dry the chip. The chips were placed on lint-free Kimtech wipes (Kimberly-Clark, Dallas, TX, USA, Product No.: 7552) to allow the remaining water to drain. The chips were then transferred to the sample stage for the chips for fluorescent and interferometric imaging.

### 4.4. Statistical Methods

The depicted results show the mean measurement of three independent biological replicates and their respective technical triplicates ± standard error of mean, unless stated otherwise. Graphs were generated using Graphpad Prism 6 software. Fluorescent tetraspanin counts and interferometric measurements were directly exported from ExoView Analyzer. 

For the fluorescent tetraspanin colocalization analysis, the data were exported assuming that the particles trapped on the anti-tetraspanin spots were positive for the respective tetraspanin regardless of the fluorescence signal. The simulated tetraspanin distribution resembles the theoretical random distribution of the measured signals. The predicted probability of a combination of tetraspanin makers was determined by multiplying the individual tetraspanin-positive sEVs divided by the total number of sEV detected to the power of three (Formula (1)). For non-colocalizing or double-positive sEV, the number of sEV not carrying the respective tetraspanin was used to determine the probability of colocalization.
(1)$$P_{\left(\mathrm{CD}9\left|\mathrm{CD}63\right|\mathrm{CD}81\right)}=\frac{\#\mathrm{CD}9_{\mathrm{pos}}*\#\mathrm{CD}63_{\mathrm{pos}}*\#\mathrm{CD}81_{\mathrm{pos}}}{{\left(\#\mathrm{total}\;\mathrm{vesicles}\right)}^{3}}$$

This probability was used to simulate the random reassignment of the signals to particles (according to their individual probability/frequency). Appendix A shows an exemplary calculation. This assessment of randomness enables us to determine if certain observed colocalization events are over- or under-represented. 

## Figures and Tables

**Figure 1 ijms-23-08544-f001:**
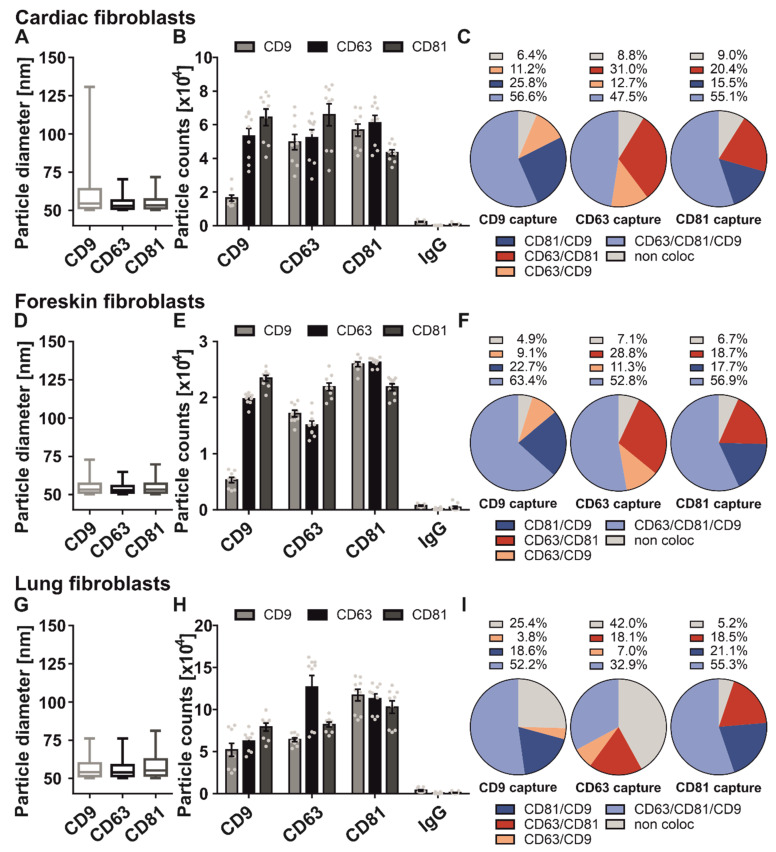
Comparison of tetraspanin composition and characteristics of sEV derived from fibroblasts of different origins. SEV derived from cardiac fibroblasts (**A**–**C**), foreskin fibroblasts (**D**–**F**) and lung fibroblasts (**G**–**I**) were captured using specific antibody-coated spots against CD63, CD81, and CD9 and analyzed using the ExoView R100 platform. (**A**,**D**,**G**) Size distribution of fibroblast-derived sEV on tetraspanin capture spots analyzed using the SP-IRIS mode of the ExoView R100 platform. Box and Whisker Plot depicts the size distribution of all measured sEV on their respective capture spots. Box indicates the 25 to 50 percentile range and whiskers indicate the 5 to 95 percentile range. (**B**,**E**,**H**) Total fluorescent particle count of fibroblast-derived sEV on tetraspanin capture spots and mouse IgG isotype control analyzed using fluorescent antibodies against CD9/CD63/CD81. Plotted is the mean ± SEM of three independent biological experiments with three technical replicates, respectively. Single measurements are indicated as dots (**C**,**F**,**I**) Tetraspanin colocalization (coloc) of fibroblast-derived sEV using three fluorescent channels and overlay of fluorescent images. Data shown represents the respective tetraspanin colocalization fraction [%] out of all detected sEV. Plotted is the mean of three independent biological with three technical replicates, respectively.

**Figure 2 ijms-23-08544-f002:**
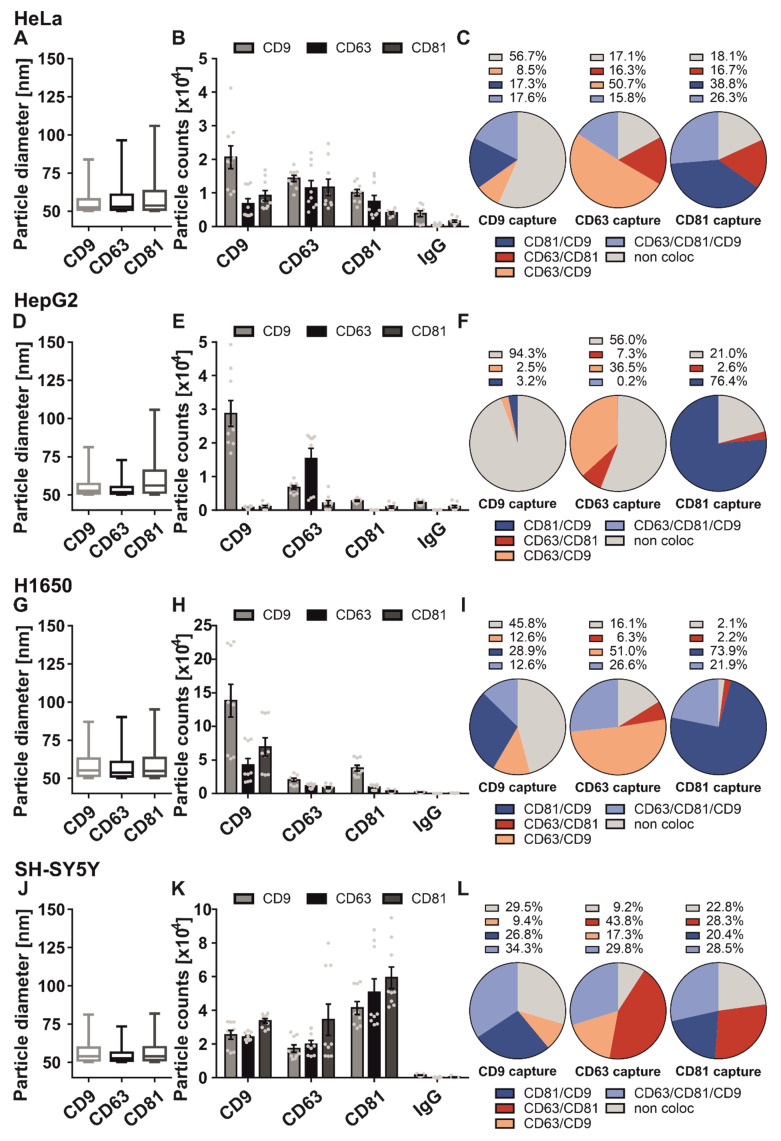
Comparison of tetraspanin composition and characteristics of sEV derived from different cancer cells. SEV derived from HeLa cells (**A**–**C**), HepG2 cells (**D**–**F**), H1650 cells (**G**–**I**) and SH-SY5Y cells (**J**–**L**) were captured using specific antibody-coated spots against CD63, CD81, CD9 and analyzed using the ExoView R100 platform. (**A**,**D**,**G**,**J**) Size distribution of cancer cell-derived sEV on tetraspanin capture spots analyzed using the SP-IRIS mode of the ExoView R100 platform. Box and Whisker Plot depicts the size distribution of all measured sEV on their respective capture spots. Box indicates the 25 to 50 percentile range and whiskers indicate the 5 to 95 percentile range. (**B**,**E**,**H**,**K**) Total fluorescent particle count of cancer cell-derived sEV on tetraspanin capture spots and mouse IgG isotype control analyzed using fluorescent antibodies against CD9/CD63/CD81. Plotted is the mean ± SEM of three independent biological experiments with three technical replicates, respectively. Single measurements are indicated as dots (**C**,**F**,**I**,**L**) Tetraspanin colocalization (coloc) of cancer cell-derived sEV using three fluorescent channels and overlay of fluorescent images. Data shown represents the respective tetraspanin colocalization fraction [%] out of all detected sEV. Plotted is the mean of three independent biological with three technical replicates, respectively.

**Figure 3 ijms-23-08544-f003:**
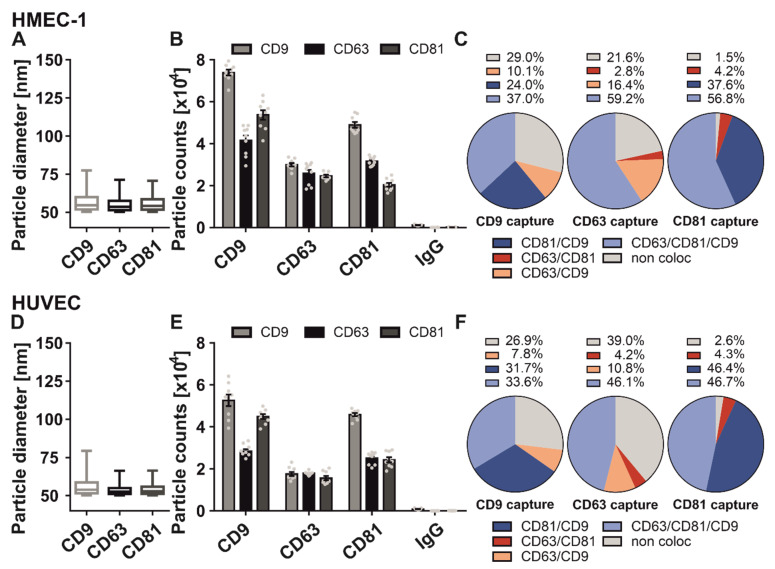
Comparison of tetraspanin composition and characteristics of sEV derived from different endothelial cells. SEV derived from HMEC-1 cells (**A**–**C**) and HUVEC cells (**D**–**F**) were captured using specific antibody-coated spots against CD63, CD81, CD9 and analyzed using the ExoView R100 platform. (**A**,**D**) Size distribution of endothelial cell-derived sEV on tetraspanin capture spots analyzed using the SP-IRIS mode of the ExoView R100 platform. Box and Whisker Plot depicts the size distribution of all measured sEV on their respective capture spots. Box indicates the 25 to 50 percentile range and whiskers indicate the 5 to 95 percentile range. (**B**,**E**) Total fluorescent particle count of endothelial cell-derived sEV on tetraspanin capture spots and mouse IgG isotype control analyzed using fluorescent antibodies against CD9/CD63/CD81. Plotted is the mean ± SEM of three independent biological experiments with three technical replicates, respectively. Single measurements are indicated as dots (**C**,**F**) Tetraspanin colocalization (coloc) of endothelial cell-derived sEV using three fluorescent channels and overlay of fluorescent images. Data shown represents the respective tetraspanin colocalization fraction [%] out of all detected sEV. Plotted is the mean of three independent biological with three technical replicates, respectively.

**Figure 4 ijms-23-08544-f004:**
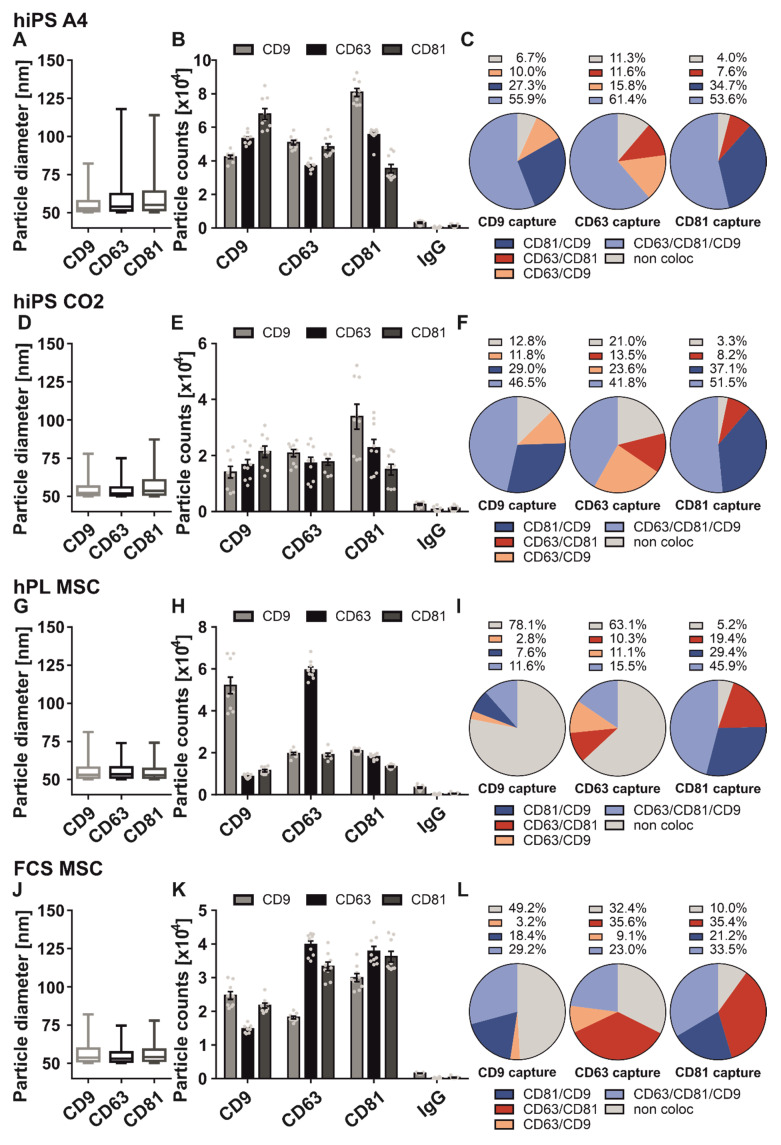
Comparison of tetraspanin composition and characteristics of sEV derived from different stem cells. SEV derived from hiPS A4 (**A**–**C**), hiPS CO2 (**D**–**F**), MSC cultured in hPL containing media (**G**–**I**) and MSC cultured in FCS containing media (**J**–**L**) were captured using specific antibody-coated spots against CD9, CD63, CD81, and analyzed using the ExoView R100 platform. (**A**,**D**,**G**) Size distribution of stem cell-derived sEV on tetraspanin capture spots analyzed using the SP-IRIS mode of the ExoView R100 platform. Box and Whisker Plot depicts the size distribution of all measured sEV on their respective capture spots. Box indicates the 25 to 50 percentile range and whiskers indicate the 5 to 95 percentile range. (**B**,**E**,**H**) Total fluorescent particle count of stem cell-derived sEV on tetraspanin capture spots and mouse IgG isotype control analyzed using fluorescent antibodies against CD9/CD63/CD81. Plotted is the mean ± SEM of three independent biological experiments with three technical replicates, respectively. Single measurements are indicated as dots (**C**,**F**,**I**) Tetraspanin colocalization (coloc) of stem cell-derived sEV using three fluorescent channels and overlay of fluorescent images. Data shown represents the respective tetraspanin colocalization fraction [%] out of all detected sEV. Plotted is the mean of three independent biological with three technical replicates, respectively.

**Figure 5 ijms-23-08544-f005:**
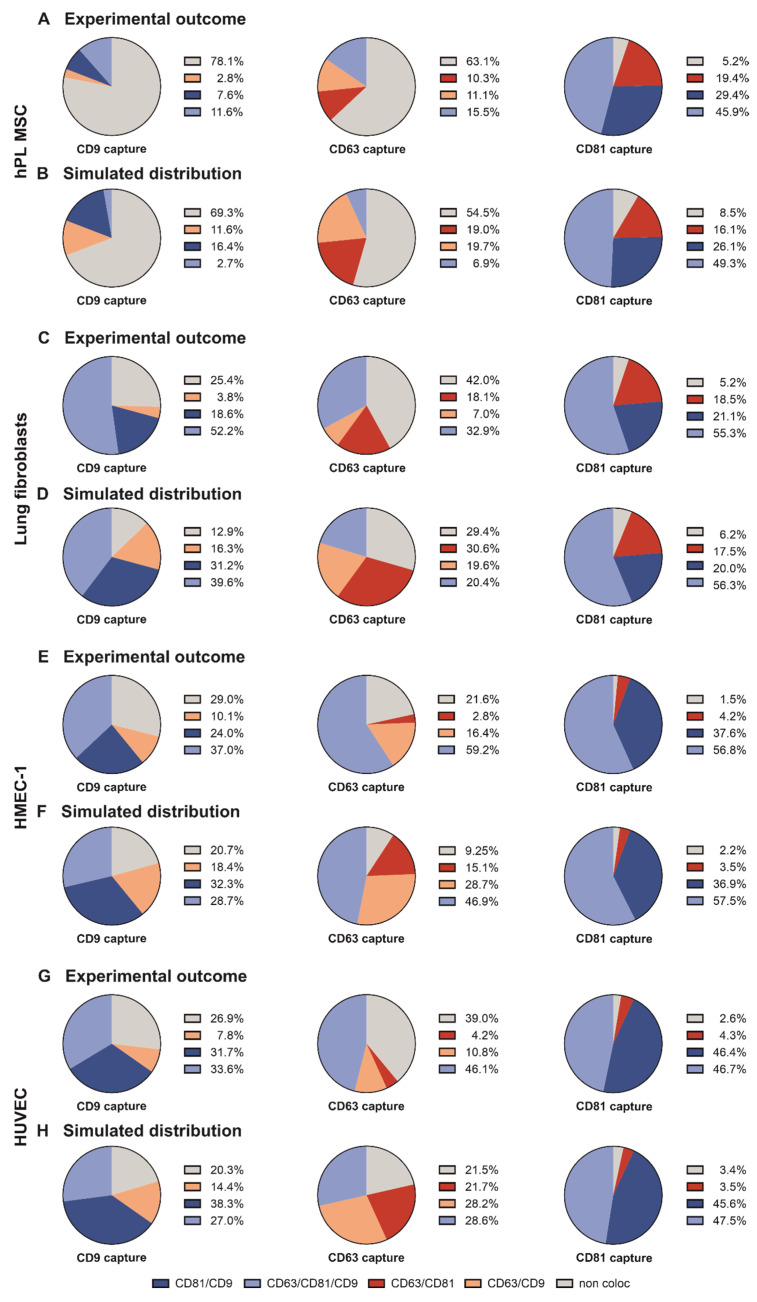
SEV derived from different cell lines differ in their tetraspanin colocalization from the predicted random distribution. Shown is the experimental outcome of tetraspanin colocalization (coloc) (**A**,**C**,**E**,**G**) on the CD9/CD63/CD81 capture spots as seen above and the predicted outcome under the assumption that tetraspanins are randomly distributed in sEV (**B**,**D**,**F**,**H**). Shown data refers to sEV populations from HMEC-1 (**A**,**B**), HUVEC (**C**,**D**), MSC (**E**,**F**) and HFL1 (**G**,**H**). Experimental data shown represents the respective tetraspanin colocalization fraction [%] out of all detected sEV. Plotted is the mean of two independent biological with three technical replicates, respectively. The predicted random distribution was calculated from the respective experimental data and was generated as indicated.

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
