# Peer review of "Detailed Characterization of Small Extracellular Vesicles from Different Cell Types Based on Tetraspanin Composition by ExoView R100 Platform"

_ijms, 2022, doi:10.3390/ijms23158544_

Round 1
Reviewer 1 Report
The title of submitted manuscript is very interesting. The reader would expect deep characterization and comparison of sEVs from different cell types. Instead of that it is complicated text about size distribution and tetraspanins composition of sEVs. Figures are disarranged and it is not clear what the autors wanted to show. What is the real importance from obtained results? How this could be utilized?
Several points need to be addressed:
1. Please, justify why were selected for tetraspanin characterization such cell lines?
2. Authors stated, that for experiments were used cells up to 25 passages. Are there any variations in tetraspanin cargo seen between passages for every type of cell lines/cells? Please, specify with figure.
3. For the confirmation that authors are working with EVs, it is necessary to fullfill ISEV/ MISEV "minimal experimental requirements for definition of extracellular vesicles and their functions", covering EVs isolation, characterization and functional studies. In the submitted manuscript are missing infromation about EVs separation/isolation. How is EVs purity?
4. Could you please compare the results with another experiments? E.g. NTA or flow cytometry?
Author Response
Reviewer 1
We would like to take this opportunity to thank the reviewer for his assessment and his comments.
The title of submitted manuscript is very interesting. The reader would expect deep characterization and comparison of sEVs from different cell types. Instead of that it is complicated text about size distribution and tetraspanins composition of sEVs. Figures are disarranged and it is not clear what the autors wanted to show. What is the real importance from obtained results? How this could be utilized?
- The primary goal of this work is to provide information on the tetraspanin composition of different EV populations. A corresponding overview study that characterizes EV populations based on their size and CD9/CD63/CD81 composition without artificial purification does not yet exist in this form. We see this work as a library for scientists, hence the detailed description of the data we show in this paper. This information can be used for further studies investigating the physiological function of EVs or choosing a proper immune-affinity EV isolation method. We have emphasized the aim and utilization of the results in the introduction and the discussion.
- We have clarified the title and the text of the manuscript to prevent misunderstandings.
Several points need to be addressed:
- Please, justify why were selected for tetraspanin characterization such cell lines?
- To gain better insight into the complexity and heterogeneity of sEV, we created a library of sEV phenotypes secreted by different cell types, including therapeutically relevant sEV from mesenchymal stem cells (MSCs) and endothelial cells. In addition, we have the EV populations of cancer cell lines of different tumor entities, which are also used for routine work, such as HeLa cells. Our goal is to describe a relatively comprehensive overview of a wide variety of EV populations.
- Authors stated, that for experiments were used cells up to 25 passages. Are there any variations in tetraspanin cargo seen between passages for every type of cell lines/cells? Please, specify with figure.
- To include variations in tetraspanin composition, three biological replicates at different passages / from different donors were used in the study. As it can be seen from the results figures, some minor variations in tetraspanin composition can be seen but the overall profile remains consistent between replicates/donors. The explanation of how long a cell line is cultivated, for example, intends to ensure the reproducibility of the experiments.
- For the confirmation that authors are working with EVs, it is necessary to fullfill ISEV/ MISEV "minimal experimental requirements for definition of extracellular vesicles and their functions", covering EVs isolation, characterization and functional studies. In the submitted manuscript are missing infromation about EVs separation/isolation. How is EVs purity?
- The advantage of EV characterization by ExoView R100 instrument is the direct analysis of EVs in biofluids without purification of EV like ultracentrifugation. This fact is a considerable advantage over other analytical methods as it avoids artificial enrichment of the EV population during the purification process.
Therefore, we have described our analysis method and explained the corresponding advantages and disadvantages in the manuscript.
We have been able to detect the size distribution and specific EV markers in different ways using our measurement method.
- Could you please compare the results with another experiments? E.g. NTA or flow cytometry?
- A recent study by Mizenko et al (2021) compares the different methods such as flow cytometry and SP-IRIS coupled with immunofluorescence staining. See: Mizenko, Rachel R.; Brostoff, Terza; Rojalin, Tatu; Koster, Hanna J.; Swindell, Hila S.; Leiserowitz, Gary S. et al (2021): Journal of nanobiotechnology 19 (1), p. 250. DOI: 10.1186/s12951-021-00987-1.
We summarized the findings from this and other studies in the manuscript and placed our results in this context.

Reviewer 2 Report
In the manuscript "Characterization of small extracellular vesicles derived from different cell types by tetraspanin composition" by Breitweiser and others makes use of single-particle interferometric reflectance imaging sensing, utilizing the ExoView R-100 platform to examine the tetraspannin composition of EVs released from various cell types. This article contains important and useful information, but I found lack of clarity in the presentation reduces the impact of the article in its current form.
Major comment: There is a lot of data presented in the manuscript, but there is little coherence in the way the information is presented and little effort to present a framework to allow the readers to understand the data. At the heart is that different cell types produce EVs with different combinations of CD9, CD63 and CD81, but there is little effort to provide a hypothesis for why this might be. There are articles that provide data to support hypothesis. For example (Mathieu M, Névo N, Jouve M, Valenzuela JI, Maurin M, Verweij FJ, Palmulli R, Lankar D, Dingli F, Loew D, Rubinstein E, Boncompain G, Perez F, Théry C. Specificities of exosome versus small ectosome secretion revealed by live intracellular tracking of CD63 and CD9. Nat Commun. 2021 Jul 19;12(1):4389. doi: 10.1038/s41467-021-24384-2. PMID: 34282141; PMCID: PMC8289845.) provides a potential framework for the data. While I understand the authors may be reluctant to speculate overmuch, without a framework to think about the results, the article is difficult to read and process.
Other concerns
1. The authors define sEVs as EVs between 30-200 nm-in diameter. I think EVs are generally defined in the same way and sEVs are sometimes used to denote smaller EVs 30-100 nm. It would be clearer just to call the the 30-200 nm vesicles EVs.
2. Page 2.....vesicle samples contain far more particles beyond sEVs......not pure enough. Is this referencing exomeres? What does not pure enough mean?
3. According to the first three paragraphs of results, ExoView is limited to 50 nm resolution and the sizes of EVs determined were slightly above 50 nm, and this was based on light interferometry. Then the EVs were detected by immunofluorescence. TEM typically finds EVs to range from 30-200 nm, but at least in many cell types, with most in the 30-70 nm range, with 30 nm being around the theoretical smallest limit for a lipid bilayer. I am not sure how useful the detected diameters are given that ExoView probably misses many EVs that are below its detection limit.
Are only detected spots analyzed by IF or are there other spots that are ignored because they were not detected by light interferometry?
2. Results section mostly reads as a description of results but with no framework. Why should the reader care that 37% of the CD9 positive sEVs also contain CD63 and CD81? It is difficult to read.
3. hPL Is that human platelet lysate? It may be defined in the manuscript but I could not find it. It seems like the supplementation experiments should be included in the abstract.
4. Discussion First three paragraphs do not discuss data from the article, and would be more appropriate in the Introduction. Some overlaps with material already in the Introduction.
5. Reported size differs from TEM and NTA. This paragraph is not very clear. It is argued that using all three surface markers should...characterize the majority of EVs, but that is only if the majority of EVs contain at least one of the markers. Do we know this? Are there data to support this?
6. Discussion page 17. Tetraspanin changed when cell culture conditions were altered...should be considered when exploiting therapeutic value. I assume this means that EVs from a particular cell type may only be of value if the cell is grown under certain conditions. This could be stated more clearly.
7. This study would be stronger if the ExoView data were confirmed by another approach, for example by immunoprecipitation and Westerns to show expected differences in the amount of markers in particualr types of EVs. For example, CD9 IPs from HMEC-1 cells would be expected to have a lot of CD81, while CD9 IPs from hPL MSCs would be expected to have much less. At least a little data from IPs or another method would increase the confidence in the ExoView data. Another idea would be to compare the cellular distribution of CD9, CD63 and CD81 in various cells by immunofluorescence. That might give a clue for the differences in the EV compositions detected.
8. I am not sure the probability analysis is helpful. There are other variables....for example relative abundance of the tetraspanins in the cells and the odds of enough tetraspanins being packaged into an EV to make it detectable that I think must be known to use this sort of analysis to draw conclusions about the presence or absence of specific sorting pathways being involved. Maybe I am missing something.
This article contains important data derived from an innovative approach. However, because the data is presented as a descriptive, empirical, narrative, with no hypothesis or rationale for the variations in tetraspanins, it will not be very useful for readers outside of this specialty. I think that it is also needs to be tightened up. Notably, the first three paragraphs of the Discussion should be, or already is, in the Introduction.
Author Response
Reviewer 2
In the manuscript "Characterization of small extracellular vesicles derived from different cell types by tetraspanin composition" by Breitweiser and others makes use of single-particle interferometric reflectance imaging sensing, utilizing the ExoView R-100 platform to examine the tetraspannin composition of EVs released from various cell types. This article contains important and useful information, but I found lack of clarity in the presentation reduces the impact of the article in its current form.
We would like to take this opportunity to thank you for your comprehensive review of our manuscript. The constructive assessment and comments helped significantly to improve the manuscript.
Major comment: There is a lot of data presented in the manuscript, but there is little coherence in the way the information is presented and little effort to present a framework to allow the readers to understand the data. At the heart is that different cell types produce EVs with different combinations of CD9, CD63 and CD81, but there is little effort to provide a hypothesis for why this might be. There are articles that provide data to support hypothesis. For example (Mathieu M, Névo N, Jouve M, Valenzuela JI, Maurin M, Verweij FJ, Palmulli R, Lankar D, Dingli F, Loew D, Rubinstein E, Boncompain G, Perez F, Théry C. Specificities of exosome versus small ectosome secretion revealed by live intracellular tracking of CD63 and CD9. Nat Commun. 2021 Jul 19;12(1):4389. doi: 10.1038/s41467-021-24384-2 . PMID: 34282141 ; PMCID: PMC8289845. ) provides a potential framework for the data. While I understand the authors may be reluctant to speculate overmuch, without a framework to think about the results, the article is difficult to read and process.
- This study aimed to provide a comprehensive overview of sEV populations based on their tetraspanin composition without artificial enrichment. We see this work as a library for scientists working on EV biology. This is the reason why we described the results in such detail. We consider our study as a starting point for further characterization studies on sEV populations. The current and future work of our group is investigating the physiological function of the different sEV populations, but this was not the scope of this study. However, we have emphasized this aspect in the discussion part.
- Thank you very much for the hint. We have put our results in the context of the results of Mathieu et al. (2021), which provides a new perspective on our data.
Other concerns
- The authors define sEVs as EVs between 30-200 nm-in diameter. I think EVs are generally defined in the same way and sEVs are sometimes used to denote smaller EVs 30-100 nm. It would be clearer just to call the the 30-200 nm vesicles EVs.
- According to the MISEV2018 guidelines, we have defined sEV populations as vesicles < 200 nm in diameter in the manuscript.
- Page 2.....vesicle samples contain far more particles beyond sEVs......not pure enough. Is this referencing exomeres? What does not pure enough mean?
- We have changed the text to avoid misunderstanding.
- According to the first three paragraphs of results, ExoView is limited to 50 nm resolution and the sizes of EVs determined were slightly above 50 nm, and this was based on light interferometry. Then the EVs were detected by immunofluorescence. TEM typically finds EVs to range from 30-200 nm, but at least in many cell types, with most in the 30-70 nm range, with 30 nm being around the theoretical smallest limit for a lipid bilayer. I am not sure how useful the detected diameters are given that ExoView probably misses many EVs that are below its detection limit.
Are only detected spots analyzed by IF or are there other spots that are ignored because they were not detected by light interferometry?
- We pointed out the detection limit of the light interferences was 50 nm in the results. To avoid misunderstandings, we included that spots analysed in IF do not have to be positive spots in interferometry mode. Thus, particles with sufficient fluorescent signal will be detected irrespective of their determined size (by SP-IRIS).
- Results section mostly reads as a description of results but with no framework. Why should the reader care that 37% of the CD9 positive sEVs also contain CD63 and CD81? It is difficult to read.
- We see this study as a library of information for scientists, hence the detailed description of the data we show in this work. This information can be used for further studies to investigate the physiological function of EVs or to select an appropriate method to isolate EVs with immunoaffinity. We have highlighted the aim and use of the results in the Introduction and Discussion. We also changed the title to avoid misunderstanding.
- hPL Is that human platelet lysate? It may be defined in the manuscript but I could not find it. It seems like the supplementation experiments should be included in the abstract.
- Thank you very much for the hint! We explained that hPL is human platelet lysate. We defined it in the Material and Method part, but it was missing in the Results. Moreover, we have now included the findings of these experiments in the abstract.
- Discussion First three paragraphs do not discuss data from the article, and would be more appropriate in the Introduction. Some overlaps with material already in the Introduction.
- We modified the Introduction and Discussion part in a more structured and clearer way.
- Reported size differs from TEM and NTA. This paragraph is not very clear. It is argued that using all three surface markers should...characterize the majority of EVs, but that is only if the majority of EVs contain at least one of the markers. Do we know this? Are there data to support this?
- Even though there is currently no publication that directly analyzed the percentage of tetraspanin-positive (CD9,63,81) EVs in comparison to total EVs, several publications utilize the capture by tetraspanins for “total” EV capture.
ELISA: doi: 10.1371/journal.pone.0005219;
doi: 10.1002/ijc.29664
Flow Cytometry:
doi: 10.1038/s41598-019-38516-8
doi: 10.1038/s41598-017-11249-2
Chip assays:
doi: 10.1039/c9lc00624a
doi: 10.1021/acs.analchem.9b04852
Additionally, Mizenko et al. (2021) showed that the detection of non-tetraspanin proteins is similar between unspecifiable bound EVs (biotinylated) and tetraspanin bound vesicles.
This indicates that in this case, the majority of EVs are positive for at least one of the three tetraspanins.
- Discussion page 17. Tetraspanin changed when cell culture conditions were altered...should be considered when exploiting therapeutic value. I assume this means that EVs from a particular cell type may only be of value if the cell is grown under certain conditions. This could be stated more clearly.
- Thank you very much for the hint. We rewrite this topic in the Discussion and abstract.
- This study would be stronger if the ExoView data were confirmed by another approach, for example by immunoprecipitation and Westerns to show expected differences in the amount of markers in particualr types of EVs. For example, CD9 IPs from HMEC-1 cells would be expected to have a lot of CD81, while CD9 IPs from hPL MSCs would be expected to have much less. At least a little data from IPs or another method would increase the confidence in the ExoView data. Another idea would be to compare the cellular distribution of CD9, CD63 and CD81 in various cells by immunofluorescence. That might give a clue for the differences in the EV compositions detected.
- We agree that additional experiments with other detection approaches would in theory improve/support the findings from the ExoView platform. The problem that we see is that there is no suitable alternative method that allows for the detection on single-EV level without requiring isolation/purification. Western Blot for example would require a purification/concentration step and would only allow for the detection of tetraspanins in EV bulk, therefore limiting the usefulness.
The ExoView platform has been compared to other methods elsewhere using isolated EV samples.
- doi: 10.1080/20013078.2019.1596016
- doi: 10.1186/s12951-021-00987-1
- doi: 10.3390/cells10112948
- doi: 10.1002/jev2.12079
- I am not sure the probability analysis is helpful. There are other variables....for example relative abundance of the tetraspanins in the cells and the odds of enough tetraspanins being packaged into an EV to make it detectable that I think must be known to use this sort of analysis to draw conclusions about the presence or absence of specific sorting pathways being involved. Maybe I am missing something.
- We agree that a multitude of circumstances can influence sEV packing. Our intension with the analysis of randomness was to indicate when tetraspanin colocalization in sEV populations were deviating from stochastical behavior. It was not our intention to infer that these deviations are directly linked to the presence or absence of certain sorting mechanisms. Interestingly, the populations that deviated the most from stochastical behavior possessed rather balanced tetraspanin expressions which makes it less likely that observed colocalizations deviated due to relatively low abundance of one type of tetraspanin.
